# The Influence and Action Mechanization of Mineral Mixed Material on High Fluidity Potassium Magnesium Phosphate Cement (MKPC)

**Qing Wu, Yan Zou, Jianhua Gu, Jun Xu \*, Rongjian Ji and Gang Wang**

School of Civil Engineering and Architecture, Jiangsu University of Science and Technology, Zhenjiang 212003, China; wuqing@just.edu.cn (Q.W.); zy739532536@163.com (Y.Z.); gu_hua_jian@163.com (J.G.); jirongjian123@163.com (R.J.); wgstarsky@163.com (G.W.)
\* Correspondence: xujun@just.edu.cn

**Abstract:** Potassium magnesium phosphate cement (MKPC) is a type of chemically bonded ceramic material that has higher performance compared to traditional Portland cement. To develop the spraying and crack pouring process of MKPC, the mechanical properties, volume deformation, hydration temperature, and water stability of the high-fluidity MKPC with different mineral mixed materials and their influence laws were studied. The effects of phase composition and micromorphology of hydration products on the properties of MKPC and its mechanism were analyzed using X-ray diffraction (XRD), thermogravimetry/differential thermal analysis (TG/DTA), and scanning electron microscopy (SEM). The results show that fly ash and metakaolin will not reduce the fluidity of MKPC paste because of their material properties, and silica fume will reduce the fluidity of MKPC paste because of its large specific surface area and high water absorption. Metakaolin can react with phosphate to form aluminum phosphate gel and fill the pores between the crystals because it has a higher activity, which can significantly improve its compressive strength. However, during the later stage of hydration, there will be slight expansion, which would reduce its bonding flexural strength. The MKPC-hardened paste mixed with silicon ash has optimal stability: therefore, it has the highest bonding flexural strength. Microcosmic analysis shows that mineral mixed material plays a physical filling role and participates in the hydration reaction as an active ingredient to improve the early hydration degree, which can change the crystal size and micromorphology of MKPC-hardened paste and make the structure more compact.

**Keywords:** potassium magnesium phosphate cement (MKPC); high fluidity; mineral mixed material; microcosmic analysis

## 1. Introduction

Potassium magnesium phosphate cement (Potassium magnesium phosphate cement, MgKPO4·6H2O, code MKPC) is a new type of inorganic cementing material that possess advantages such as quick hardness and early strength, strong adaptability to environmental temperature, high bond strength with old concrete, small volume deformation, and resistance to abrasion, frost, and salt frost erosion [1–4]. Moreover, because of the hydration mechanism and the particularity of cement raw materials, MKPC shows excellent protection for reinforcement [5,6] and shows very good compatibility with fibers [7–9], and has considerable development potential in marine concrete [10,11]. Currently, MKPC is primarily used as a fast repair material and other its applications in engineering are rarely reported. Recently, the Eon Coat Company from USA developed one phosphate cement-based anticorrosion coating product [12], which can ensure that the base material exposed for 1000 h in salt

fog chamber regulated by ASTM standard does not undergo corrosion or penetration. Moreover, because the use of prefabricated building is increasing, ensuring the MKPC base material as the joint material of concrete precast components has an obvious economic and environmental protection benefits [13].

Generally, the workability of mortar and concrete primarily includes fluidity, cohesiveness, and water retention. Fluidity is the most important and convenient technical indicator to determine the quality of workability, and it has important influence on uniformity and molding difficulty of the mixture, thus further affecting the strength and durability of hardened paste. Wang et al. [14] reported that fluidity has a very high sensitivity to raw material particle size, cement-sand ratio, and water–binder ratio of magnesium phosphate cement (MPC). As a mixed material, the fly ash can play a good role of reducing water and plasticization. Therefore, Wang et al. [15] prepared a compound water reducer, which could considerably increase the fluidity and improve the strength of MPC. Lin et al. [16] reported that the water reducer produced by graphene oxide was able to considerably improve the fluidity of MPC, and was not affected by the acidic environment of the system. Sun et al. [17] reported that the aggregate ratio and aggregate size are the primary factors that affect the self-leveling of the MPC paste by studying the effect of aggregate on MPC mortar. However, there have been reports on systematic studies concerning high fluidity MKPC mortar and its performance are rare.

We developed a composite retarder for MKPC during the early period [18], which can be used to freely adjust and control the condensation time of MKPC between 20 and 180 min and establishes the foundation for extensive application of the retarder. Thus, in this study, we examined the trial mix of MKPC mortar with high fluidity through a large quantity of trial and mineral materials that were mixed to improve characteristics. Furthermore, we examined the influence of mineral mixed materials to fluidity, strength, volume deformation, hydration temperature, and water stability of MKPC mortar with high fluidity, as well as conducted the mechanistic analysis through related physical and chemical performance test and microcosmic analysis.

## 2. Materials and Methods

### 2.1. Raw Materials

The prepared MKPC mortar consists of dead burned magnesium oxide ($M_gO$) powder, potassium hydrogen phosphate ($KH_2PO_4$; KDP), and a composite retarder (CR). $M_gO$ with a specific surface area of 216 $m^2$/kg, average particle size of 60.06 μm, (10) and specific gravity of 3.11 g/$cm^3$ was produced by Magnesium Sand Plant of Dongfanghong Hydropower Station in Hengren, Benxi, China. We obtained industrial grade KDP from Lianyungang Gree Chemical, Lianyungang, China. The primary particle size ranged from 245 to 350 μm, and the content of effective components was 98%. CR was prepared using borax ($Na_2B_4O_7 \cdot 10H_2O$), disodium hydrogen phosphate dodecahydrate ($Na_2HPO_4 \cdot 12H_2O$), and some inorganic salts in a certain proportion. Other materials include chemical additives such as water glass (WG; $Na_2SiO_3$ solution, 33% effective content), mineral mixed material, and fine aggregates. Mineral mixed materials include fly ash (FA; average particle size = 45.12 μm), silica fumes (SF; particle size = ~0.01–0.5 μm), and metakaolin (MK; average particle size = ~10 μm). The fine aggregates are the medium sands by filtering sand with particle size of >2.36 mm, and the modulus of fineness is 2.2. The primary components of various powders are listed in Table 1.

**Table 1.** The chemical components of raw material powders (Mass fraction, %).

| Mineral Admixture | $AL_2O_3$ | $S_iO_2$ | $C_aO$ | $M_gO$ | $F_{e2}O_3$ | $T_iO_2$ | $N_{a2}O$ | $K_2O$ | Others |
|---|---|---|---|---|---|---|---|---|---|
| MO | 0.49 | 3.76 | 2.33 | 92.58 | 0.67 | 0.02 | | | 0.15 |
| SF | 0.44 | 93.16 | 0.44 | 1.37 | 0.27 | | 1.3 | | 3.02 |
| MK | 46.12 | 48.73 | 0.12 | 0.13 | 0.35 | 1.32 | | 0.1 | 3.13 |
| FA | 28.33 | 49.20 | 4.87 | 2.17 | 6.78 | 1.20 | 1.27 | 0.65 | 5.53 |

### 2.2. Preparation and Maintenance of Specimen

In this study, we examined four matching ratios of the MKPC mortar. According to the test result during the early period and relative references [19–21], To ensure sufficient paste thickness between sand particles and molding time, the cement–sand ratio was set to 1:0.75. In order to improve the fluidity of the MKPC slurry as much as possible, the water binder ratio is determined by the maximum water consumption of the strength of the slurry under the premise of no water glass addition, which is 0.18 through test verification. Furthermore, the mineral mixed materials are internal addition rather than $M_gO$ of 10% (mass fraction), The blank group M0 mix ratio is shown in Table 2.

**Table 2.** M0 mix ratio of blank group.

| | m(KDP)/m($M_gO$) | m(CR)/m($M_gO$) | m(w)/m(MKPC) | m(WG)/m(MKPC) | Mortar Ratio |
|---|---|---|---|---|---|
| M0 | 2:3 | 0.14 | 0.18 | 0.02 | 1:0.75 |

The specimen has two sizes: ① prism with 40 mm × 40 mm × 160 mm, which was primarily used in test compression and bending bond strength; ② prism with 25 mm × 25 mm × 280 mm, which was primarily used to test the shrinkage deformation of the hardened paste of MKPC mortar.

First, all the crystals were poured into the agitator kettle, and a small amount of water (mix the WG with water beforehand) was added, followed by stirring for 1 min, and then poured $M_gO$ , mixed materials, and the remaining water. Then, the solution was quickly stirred for 1 min, the aggregate was poured, and then quickly stirred for 2–3 min until the mixture was even. To prevent the MKPC paste by absorbing more mechanical energy and accelerating condensation, the agitation time should not be considerably long. Finally, the mixed paste was poured into the corresponding test mold, the surface was scraped, and covered using a plastic wrap to prevent water loss, and then the mold was removed after 5 h. The maintenance conditions of the specimen are divided into two types. (1) Natural curing: Place the test block after form removal at a temperature of 20 ± 2 °C, and then maintain up to a corresponding age in a curing room with a humidity of 60% ± 5%. (2) Water curing: Immerse the test specimen after 1 day of natural curing in water, change the water every 3 days, and then immersed till the corresponding aging.

### 2.3. Test Method

The cement mortar fluidity tester was adopted to determine the paste fluidity of MKPC. The cement mortar fluidity tester was used to measure the fluidity of the MKPC slurry. The mixed slurry was loaded into the test mold at one time. After scraping off the excess floating slurry, the test mold was quickly lifted for testing. The test process should be completed within 2 min. Then, the strength determination, which was performed with YZH-300.10 Constant Loading Cement Flexural and Compressive Testing Machine, was referenced to the standard GB/T 17671-1999. The test method for the bonding flexural strength of MKPC paste and ordinary Portland cement mortar formed an ordinary silicate cement mortar (cement: water: sand = 1:3.19:06; 28-day compressive strength: 34.2 MPa). After curing for 28 days, the specimen was sawn off from the middle and placed into the original test mold, and then cast into the MKPC mortar to test its bonding flexural strength. The index of specimen shrinkage deformation was represented by shrinkage ratio, and the execution was referenced to standard JC/T 603-2004. Because the condensation time of MKPC paste is extremely short [22], the false setting phenomenon of restoring plastic state exists led by continuous agitation after the initial setting. It is impossible to record truthfully using a Vicat instrument, so the hydration exothermic temperature of the slurry is used to characterize it. When the system temperature no longer increases, it is the final condensation.

The samples used for microcosmic analysis were selected from the strength test at the age of 28 days. Water curing specimen samples were obtained from the epidermis. A portion of the samples was ground into powder, passed through a 200-mesh sieve, and then stored in sealed bags. Before the

test, the samples were dried to a constant weight at 60 °C. We used a Nova Nano SEM450 type scanning electron microscope to observe the morphology of MKPC-hardened hydration outputs. Moreover, we used the D/max-RB type X-ray diffraction instrument to determine the physical phase constitution of MKPC mortar. The thermogravimetry/differential thermal analysis (TG/DTA) were performed using NETZSCH STA 409 PC/PG type thermal analyzer. The heating rate was 10 °C/min from 25 °C to 1000 °C with nitrogen as the protection gas.

## 3. Results and Discussion

### 3.1. The Influence of Mineral Mixed Materials to the Fluidity of Potassium Magnesium Phosphate Cement Mortar

The fluidity of the mortar is shown in Table 3. Because the table diameter of cement mortar fluidity tester was 300 mm, when the expanding diameter of fresh paste exceeds that of the round table surface, the recorded fluidity value was "300+ mm." Thus, the data in the table confirmed that adding a mineral mixed material powder with a larger specific surface area ratio into the MKPC mortar hydration system with a high water-binder ratio caused the paste fluidity to decrease.

**Table 3.** Mixing ratio and fluidity of potassium magnesium phosphate cement mortar.

| Code | Fluidity/mm |
|------|-------------|
| M0   | 300+ |
| M-SF | 270 |
| M-MK | 300+ |
| M-FA | 300 |

The high water-binder ratio of the paste in this experiment did not lead to bleeding and segregation because of the hydrogen bond association of ions in the WG. We will report the influence of WG on MKPC mortar in other articles due to the length limit [23]. Among the three mixed materials, the paste mixed with MK had the highest fluidity. After the addition, the paste was in the state of self-leveling, the primary reason for this was that $M_gO$ and KDP could react with both $S_i0_2$ and $Al_2O_3$ in MK with high activity and form a fluid gel substance after dissolution under the condition of high water-binder ratio, which has a certain lubrication effect; the FA with spherical particles has "ball bearing" effect and lesser influence on the fluidity of the MKPC mortar [24]. The SF with the same smooth surface had a higher water requirement and shows an obvious loss in fluidity because of its larger specific surface area.

### 3.2. The Influence of Mineral Mixed Materials to Strength of Potassium Magnesium Phosphate Cement Mortar

Figure 1 shows each matching ratio's strength change of each age under natural condition. From the perspective of compressive strength, each matching ratio had a higher early strength compared to the blank. The strength of M2 mixed with MK increases most obviously: however, the M1 mixed with SF has the highest early strength but the final strength is lesser than that of M0. At the bending bond strength, although the strength of M1 mixed with SF stably increases with curing age, other matching ratios show a back-shrinkage phenomenon. Note that this phenomenon is most significant in the blank group.

The effect of mineral mixed materials to mechanical performance of MKPC mortar is primarily the physical filling effect of micro-aggregates and the chemical effect of the participating hydration reaction as active components [25,26]. Similar to the fine aggregates, the mineral mixed materials with smaller average particle size fill the pores, increase in the compactness degree of the hardened paste, and the particle formed via the high temperature calcination has higher strength and better bonding with hydration substances, which can obviously improve the compressive strength. In the three mineral-mixed materials, the particle size of SF is the smallest, and the filling effect at the early

hydration stage is the most obvious. However, in addition the hydration reaction, micro-aggregate effect gradually weakens, and SF is difficult to be activated in an acidic environment, causing too low final strength. MK, after high temperature calcination, is a mineral mixed material with high activity. MK and FA calcined at high temperature are a kind of highly active mineral admixture. Under acid environment, $Al_2O_3$ easily reacts with phosphate to form aluminum phosphate gel to fill pores and increase strength. In addition, because the heterogeneous nucleation effect of mixed material particles, MKP forms the crystal nucleus on the particle surface with priority and delays the formation speed of MKP on the surface of $M_gO$, which favors the spread of $Mg^{2+}$ on the surface of $M_gO$ and accelerates the hydration reaction, as well as increases the hydration reaction of paste at an early stage. The dispersion effect of mineral mixed materials facilitate the reaction between $M_gO$ and KDP to a certain degree, which helps improve the early strength. From the perspective of the bending bond strength, because the occurrence of harden paste shrinkage leads to micro-stress on the connection interface, it leads to shrinkage in strength [27]. The bond performance of the MKPC mortar after adding the mineral mixed materials shows some obvious improvement, increasing the strength and reducing back shrinkage because the mineral mixed materials improve the volume stability of the hardened paste, and then the shrinkage micro-stress is reduced at the connection interface, which improves the strength. However, the mineral mixed materials ensure the early hydration of MKPC paste more sufficient and reduce the quantity of low-binding hydration products; note that the latter will absorb the water in air and transform to MKP, thus reducing its strength. The volume stability of paste mixed with SF is the best; therefore, the bending bond strength is the highest.

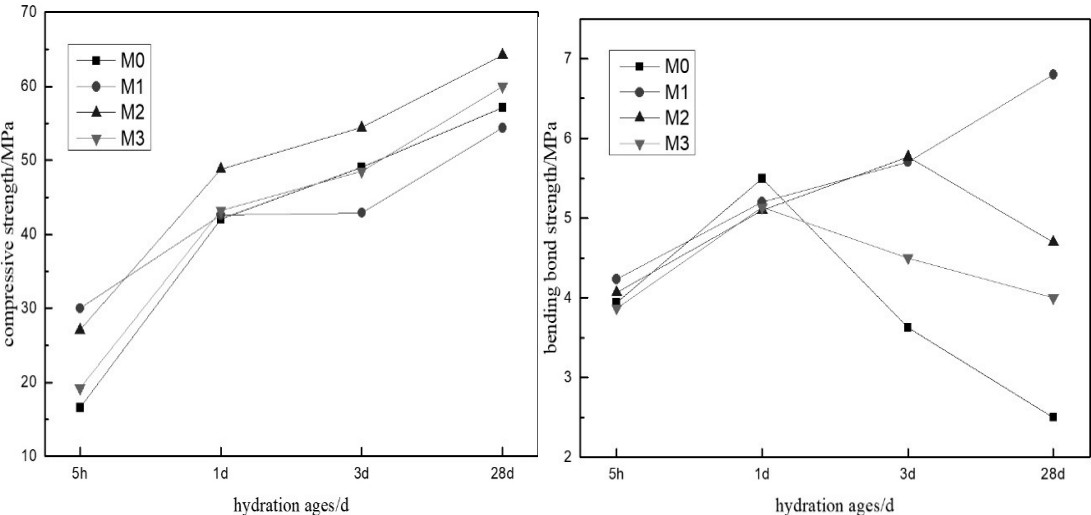

**Figure 1.** Compressive strength and bending bond strength of potassium magnesium phosphate cement (MKPC) mortar with mineral mixed materials.

### 3.3. The Influence of Mineral Mixed Materials on Volume of Potassium Magnesium Phosphate Cement Mortar

Figure 2 shows the volume deformation of four pastes within 60 days of natural curing. The figure shows that the dry shrinkage of MKPC mortar after mixed mineral materials obviously reduced and the volume stability improved. Therefore, the volume stability of the paste mixed with SF was the best.

In the raw materials of MKPC, the density of MgO powder is 3.45 g/cm$^3$, the KDP is 2.338 g/cm$^3$, and the relative density of MKP is 1.71. Therefore, the volume expansion of mortar hardening body occurs in the early stage, and the contraction occurs after 1d. The dry shrinkage of mortar blocks is mainly caused by the evaporation of capillary water and the physical–chemical combined water that functions to dissolve and diffuse cement particles. The latter mainly includes adsorbed water and interlayer water. The density of main hydration product $M_gKPO_4 \cdot 6H_2O$ (MKP) is lower than that of $M_gO$ powder. Therefore, the volume will increase with the hydration process, and the incorporated mineral admixture makes the hydration reaction more sufficient and strengthens the expansion. One

part adjusts the pore structure of the hardened body, one part compensates the tensile stress of the capillary wall due to dehydration vacuum, and the other part appears as a volume expansion [28]. In this test, a blend with a smaller average particle diameter size was used as a micro-aggregate to fill between the larger-sized $M_gO$ particles and the hydration product MKP, and form a dense gradation with a sufficient amount of unhydrated $M_gO$ particles, Reduced water evaporation. The primary component of SF was $S_iO_2$, which did not participate in the hydration reaction and filled the pores in the hardened paste in the form of particles. Furthermore, volume deformation did not occur and resisted deformation, while MK and FA were obtained because of a secondary hydration reaction. Note that the amount of products reflects the deformation of the volume. FA improved the dispersion of $M_gO$ particles in the paste, thus increasing the contact area of acid and alkali components, improving the reaction degree, and offsetting the shrinkage deformation after hardening. From Table 1 and Figure 4, because MK contains a large amount of $Al_2O_3$, which reacts with phosphate to form $AlPO_4 \cdot 2H_2O$ (having a density lower than that of $M_gO$), M2 expands; thus the bending bond strength on day 28 decreases.

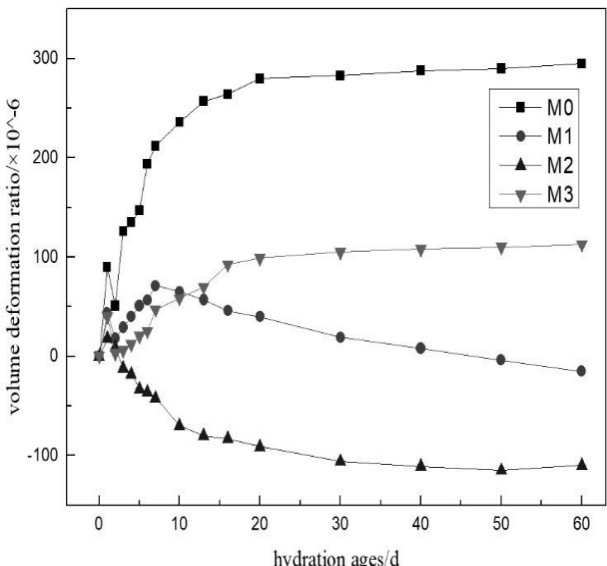

**Figure 2.** Volume deformation of MKPC hardened paste mixed with mineral materials.

*3.4. The Influence of Mineral Mixed Materials to Hydration Temperature of Potassium Magnesium Phosphate Cement Mortar*

Figure 3 shows the change in temperature of the hydration reaction of MKPC mortar under different matching ratios. The figure shows that the exothermic temperature curves of each matching ratio basically have the same morphology with two exothermic peaks and a latency. Since MgO particles are dissolved in water under acidic conditions to generate magnesium ions and heat, the average temperature of the test piece reached a peak of 37 °C at 85 min; Due to the neutralization reaction of magnesium ions and phosphate ions, the average heat of the test piece was released. The temperature reached a peak of 48 °C in about 250 min. The two kinds of temperature peaks appeared at different times, and the highest temperature that the system could reach was different.

The hydration temperature curve of slurry can reflect the speed of hydration reaction. After adding mineral admixture, the raw material is relatively loose and porous, and the volume increases. After mixing the slurry with water, more water is needed for wetting. Furthermore, additional moisture was required after adding water and mixing, which caused the water that participated in the reaction reduced, the dissolvent reduced, the solute concentration increased, and the reaction speeded up; thus, the heat release of the reaction was brought forward. The solubility of $C_aO$ contained in FA in M3 was higher than that of $M_gO$. The initial reaction with phosphate released a large amount of heat; therefore,

the temperature peak of M3 was the highest. The SF in M1 had no active materials which participated in the hydration reaction: it simply advanced the reaction temperature peak, but did not change the highest temperature of the system.

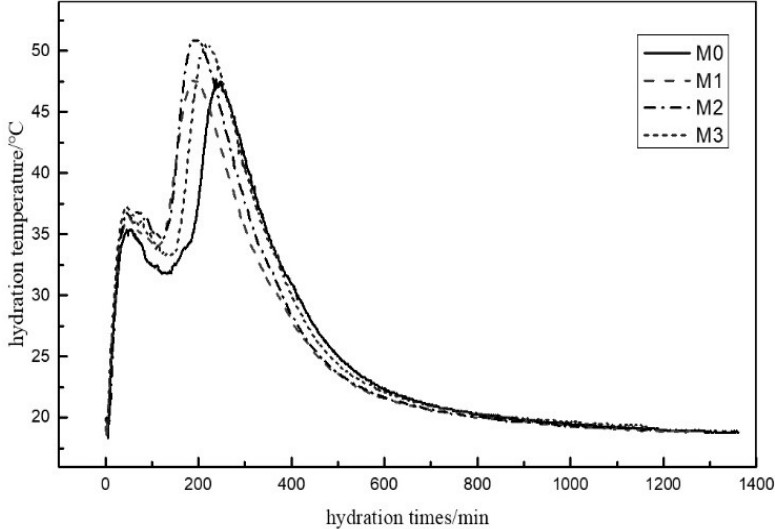

**Figure 3.** Hydration temperature of MKPC mortar with mineral admixture.

*3.5. The Influence of Mineral Mixed Materials to Water Stability of Potassium Magnesium Phosphate Cement Mortar*

Table 4 shows the water curing strength and corresponding strength residual ratio of the four different types of pastes. According to the column of compressive strength, the water resistance of the hardened paste can be improved by adding the mineral mixed materials. In terms of bending bond strength, the strength of the specimen mixed with SF after immersion was relatively stable.

**Table 4.** The short-term water maintains strength and strength residual ratio of MKPC mortar mixed with mineral mixed material.

| Code | Strength (MP$_a$) and Strength Residual Ratio | |
|------|-----------------------------|-------------------------|
|      | **Compressive Strength** | **Bending bond Strength** |
| M0 | 48.2/84.4% | 2.2/88.0% |
| M1 | 49.4/90.8% | 6.1/89.7% |
| M2 | 56.9/88.6% | 3.1/66.0% |
| M3 | 51.8/86.3% | 4.4/110.0% |

Unlike hydraulicity cementitious materials, the strength of MKPC-hardened paste will back shrink after immersion. The water will permeate into the hardened paste along the pores, dissolving the unreacted KDP, which made the water show acidity and partly dissolved MKP, thus causing structural degradation [29]. Finally, after adding the mineral mixed materials, the strength residual ratio of each paste was increased compared with that of the reference group M0. Table 4 shows that the water stability of MKPC mortar has a significant correspondence to the particle size of the mineral mixed materials. The particle size of SF was the least, and the structure was most compact and the water stability was the best.

### 3.6. Microcosmic Analysis

### 3.6.1. X-ray Diffraction

Figure 4 shows the XRD patterns of four different matching ratios after 28 days of natural curing. The figure shows that the positions of the primary characteristic peaks of all samples is basically the same and is primarily composed of the hydration product MKP, the unhydrated $M_gO$, and the fine aggregate containing $S_iO_2$. In the blank group, the diffraction peak strength of $M_gO$ is higher, which proves that mineral mixed materials can make the hydration reaction of MKPC paste more complete. Furthermore, there is no new phase appearing in the blank group M0. The M2 had new physical phase high hydration phosphate $M_{g2}H(PO_4)_2 \cdot 15H_2O$. This material was the intermediate product of the hydration reaction of MKPC paste, which could react as reaction below and produce MKP [30]. The incomplete hydration reaction of the intermediate product led to a decrease in the M0 compressive strength. Both M2 and M3 contained Al–Si mineral mixed materials, where the active components of $Al_2O_3$ reacted with phosphate. Therefore, the characteristic peak of $AlPO_4 \cdot 2H_2O$ existed in all XRD patterns.

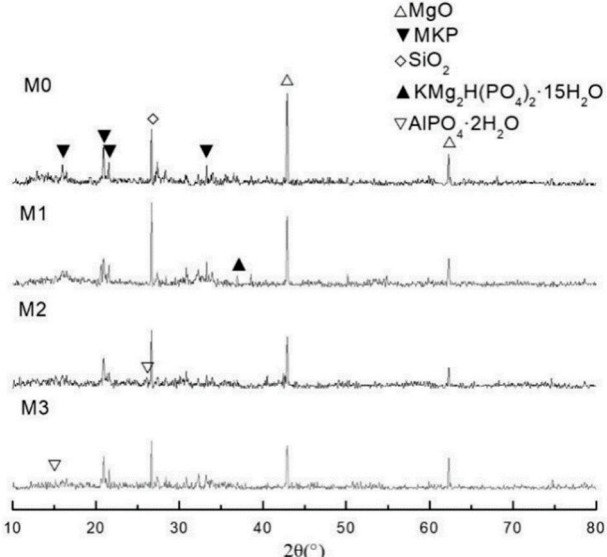

**Figure 4.** X-ray diffraction (XRD) patterns of MKPC mortar mixed with mineral mixed materials.

$$Mg_2KH(PO_4)_2 \cdot 15H_2O + K^+ + OH^- \rightarrow 2KMgPO_4 \cdot 6H_2O + 4H_2O$$

### 3.6.2. TG/DTA

Figure 5 shows the TG/DTA curves of MKPC paste at aging for 28 days under different curing conditions. The figure shows that the geometry of DTA curves of four matching ratios under two curing conditions are basically the same with an endothermic valley and a smaller exothermic peak. The endothermic valley appears at ~100 °C, which is primarily the chemical process of the removal of crystal water from the hydration product MKP. The exothermic peak appears near 380 °C, which should be the physical process of the crystal form transformation of the sample, and then DTA curve offsets to the side of heat absorption (the lower side). Now, the specimen temperature has already reached or approached to glass transition temperature (Tg), the glass transition occurred for the sample, and the heat capacity increased. Both M1 and M3 under natural curing conditions show a small exothermic peak at 70 °C, which should be the hydration reaction of condensation between active $M_gO$ and KDP without reaction after evaporation of water that hardly moved freely in the sample pores.

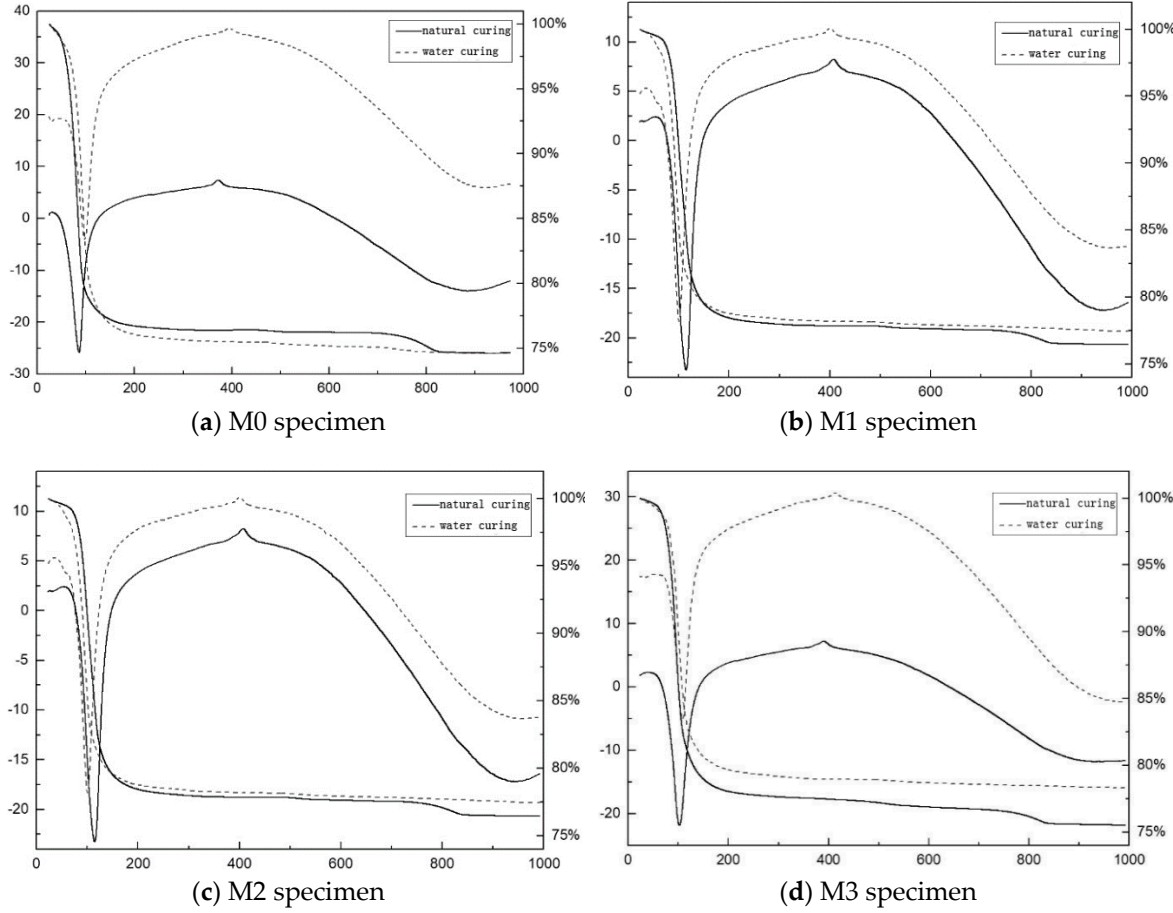

**Figure 5.** Thermogravimetry/differential thermal analysis (TG/DTA) curves of MKPC mortar under different curing conditions.

The TG curve is consistent for each ratio. When the accessories are dehydrated near 100 °C, the mass loss increases sharply, but the weight loss rate is different. The comparison of the weight loss rate at 200 °C and the mass loss rate of each matching ratio at 200 °C was shown in Table 5. It was found via the transverse comparison of mass losses of four matching ratios under two curing conditions, the blank group M0 > M1 > M3 > M2. The mineral mixed materials were utilized rather than $M_gO$, leading to the reduction of MKP content in hydration products. Therefore, the mass of specimen dehydration reduced. The test results are consistent with active components content in mineral mixed materials. Note that the active $Al_2O_3$ reacts with KDP to further consume the phosphate, reducing the MKP production and the weight loss. Because the specimens were not completely dried before testing (the specimens need to be treated by low-temperature vacuum drying), the water maintained specimen containing water was tested, which resulted in a considerable change in the mass loss rate of longitudinal comparison of different matching ratios.

**Table 5.** The mass loss of MKPC mortar specimen at 200 °C.

| Code | Natural Curing | Water Curing |
|------|----------------|--------------|
| M0 | 23.3% | 24.0% |
| M1 | 22.9% | 23.9% |
| M2 | 21.6% | 21.3% |
| M3 | 22.0% | 20.0% |

### 3.6.3. SEM

Figure 6 shows the SEM image of a 28-day hardened paste after natural curing and water curing. The figure shows that, under natural curing, MKP is a slim columnar crystal in the blank group. The different crystals overlap with each other to form a grid-like structure, and the cloud-shaped gel was wrapped between the crystals. The gel might be a hydrated magnesium silicate because of the addition of WG. Because of the effect of the mineral mixed materials, the cross-section morphology of MKPC-hardened paste changes to some extent because the SF particle size was ~0.01–0.5 μm and SF did not participate in the hydration reaction, which effectively filled the gap between crystals and the structure was more compact, thus effectively resisting shrinkage stress because of shrinkage deformation. After MK was mixed, the hydration products showed a block-like shape and the growth remained intact. There is almost no cracks on entire section, and it was closely connected and covered with many amorphous substances. Figure 6d shows that the products after FA addition show a block-like shape with fly ash particles of different sizes, filling the gaps between blocks without an obvious interface layer. Under water curing, the crystal structure was loose and porous with a coarse crystal size. A large number of irregular micro-cracks existed in the hardened paste mixed with MK and FA, which reduced the strength.

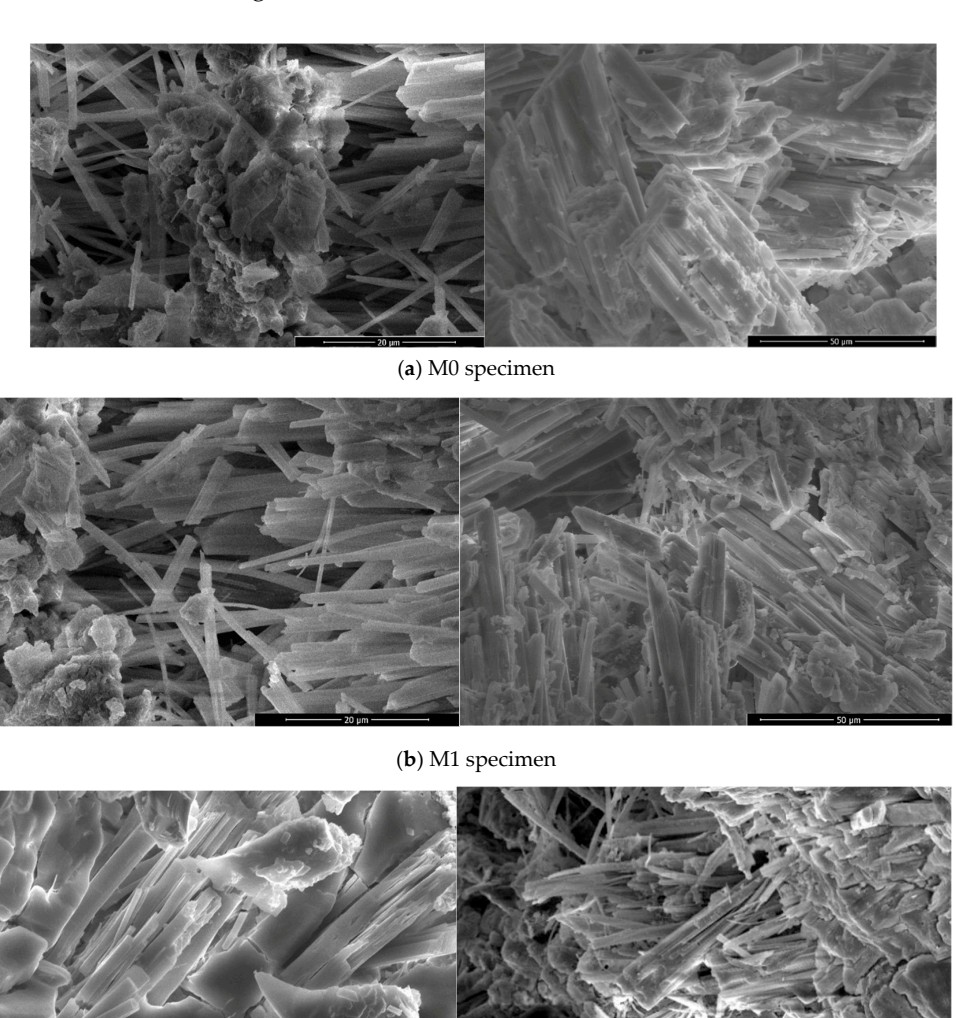

(**a**) M0 specimen

(**b**) M1 specimen

(**c**) M2 specimen

**Figure 6.** *Cont.*

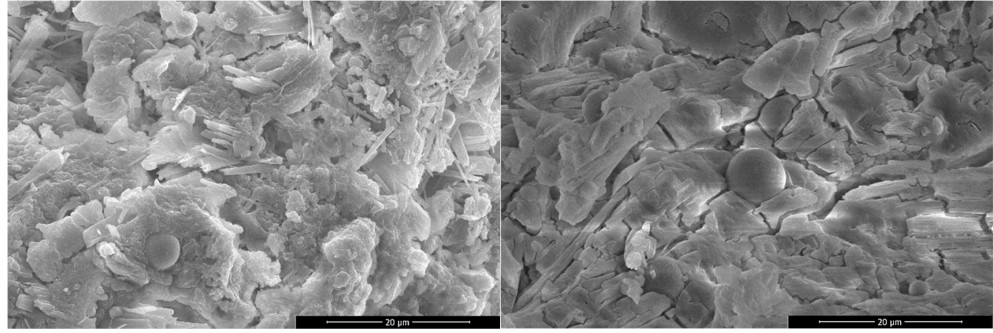

(**d**) M3 specimen

**Figure 6.** Scanning electron microscopy (SEM) image of MKPC mortar mixed with mineral mixed materials.

## 4. Conclusions

The specific surface area, micromorphology, and chemical activity of mineral mixed materials have greater influence on the fluidity of MKPC mortar. The large specific surface area and highest water demand of SF was attributed to fluidity loss. The FA of sphere particles had a morphological effect, which had lesser influence on the fluidity of MKPC mortar. Note that MK with a high activity reacted with mortar to generate a new flow state gel material, and the mortar fluidity after addition was relatively the highest and could still perform the self-leveling status.

The mineral mixed material can effectively improve the performance of the MKPC mortar with high fluidity. Furthermore, the addition of mineral mixed materials could further improve the volume stability of the hardened paste of MKPC mortar. The improvement effect of SF is the best. The hardened paste of MKPC hardly produced any shrinkage deformation, which considerably improved the bonding performance between the MKPC mortar and ordinary concrete. Because secondary hydration reaction occurred between the active $Al_2O_3$ in MK and KDP, the addition of MK was favorable for improving the compressive strength: the mineral mixed materials made the early hydration reaction more sufficient and then improved the early strength of the MKPC mortar.

The mineral mixed materials can change the micromorphology of the MKPC hydration products. Therefore, MK and FA undergo a secondary hydration reaction with KDP and generate new hydration products that block the pores. The microaggregate effect of mineral mixed materials fills the gaps of the hydration products MKP crystals, which makes the hardened paste more compact, makes the pore structure more favourable, and improves the water stability.

**Author Contributions:** Data curation, Y.Z.; Formal analysis, J.X.; Writing—original draft, Q.W.; Writing—review & editing, J.G.; Software, R.J.; Formal analysis, G.W. All authors have read and agreed to the published version of the manuscript

**Acknowledgments:** The authors gratefully acknowledge the financial support from National Key R&D Program of China (2017YFB0309904) and National Science Fund of China (51890904 and51678529).

**Conflicts of Interest:** The authors declare that they have no known competing financial interests or personal relationships that could have appeared to influence the work reported in this study.

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
