# Peer review of "The Influence and Action Mechanization of Mineral Mixed Material on High Fluidity Potassium Magnesium Phosphate Cement (MKPC)"

_jcs, doi:10.3390/jcs4010029_

Round 1
Reviewer 1 Report
The manuscript entitled " The influence and action mechanization of mineral mixed materials on high fluidity potassium magnesium phosphate cements" deals with the modification of a potassium magnesium phosphate forming cement based on MgO in order to improve the mechanical strength and chemical/ water stability of this cement. The authors used 4 different additives for the modification and investigate the chemical composition, water resistance, morphology, fluidity of the cement paste, and chemical activity.
Overall, the style of english language, in particular in section 3.3 - 3.6, makes it sometimes impossible to understand the discussion of the results. Beside this the manuscript should be revised concerning the following remarks:
1) The format of the table 2, in particular the definition of the columns is not clear.
2) Please add more details to the fuidity measurement method.
3) The method, which is used to characterize or define the condensation time is not clear. Please add more details.
4) Line 162: The authors should discuss the mechanical properties by including the XRD results. They mention, that the mechanical strength depends also on the hydration product, but they don´t correlate the specfic results in their discussion.
5) Line 186: Where is the volume stability shown? In figure 2 it is shown, that the volume deformation of M1 is higher than the one of M3.
6) Line 198: The authors mentioned the density of different chemical compounds. I recommend to add concrete values for the densities.
7) Line 202 - 206, line 209, line 229 - 231, line 292 - 293: the meaning of these sentences is not clear.
8) Line 221: The authors discuss here the temperature curve shown in figure 3. Please add the concrete peak positions in the text and refer not to them as " first" and "second" peak.
9) Table 3: Unit for strength is missing
10) line 249 - 251: The authors refer to the particle size of additives used to modify the MgO cement, but did not give information about the specific particle size values. This should be added.
11) Figure 4 and discussion of figure 4: The characteristic peak AlPO4*2H2O can not be distinguished from the underground in the diffraction pattern. How can this be explainde and how can the authors be sure, that the reaction product in the cement really exist?
12) Chinese letters in figure 5 has to be changed/removed.
13) Figure 6 (SEM picture): please improve the style of the presentation by cutting the bottom of the picture off and present the scale bar as part of the picture.
Author Response
Dear Editor,
We are very grateful to you that give us an opportunity to improve and resubmitting our manuscript. We also thank the reviewers for carefully reviewing the manuscript and providing extremely helpful feedback. We have made great efforts to revise the manuscript as possible.
Reviewers comments are numbered and appear verbatim in blue words, authors responses appear immediately following each comment in black words.
Reviewers' comments:
Reviewer #1: The manuscript entitled " The influence and action mechanization of mineral mixed materials on high fluidity potassium magnesium phosphate cements" deals with the modification of a potassium magnesium phosphate forming cement based on MgO in order to improve the mechanical strength and chemical/ water stability of this cement. The authors used 4 different additives for the modification and investigate the chemical composition, water resistance, morphology, fluidity of the cement paste, and chemical activity.
Overall, the style of english language, in particular in section 3.3 - 3.6, makes it sometimes impossible to understand the discussion of the results. Beside this the manuscript should be revised concerning the following remarks:
- The format of the table 2, in particular the definition of the columns is not clear.
Thanks.The definitions in the table have been reinterpreted and the table has been modified.(line 91)
- Please add more details to the fuidity measurement method.
Thanks. The author adds some details to the fuidity measurement method.(lines 109-111)
The mixed slurry was loaded into the test mold at one time. After scraping off the excess floating slurry, the test mold was quickly lifted for testing. The test process should be completed within 2 minutes.
- The method, which is used to characterize or define the condensation time is not clear. Please add more details.
Thanks.The author adds more details to The method, which is used to characterize or define the condensation time is not clear.(lines 121-123)
It is impossible to record truthfully using a Vicat instrument, so the hydration exothermic temperature of the slurry is used to characterize it. When the system temperature no longer increases, it is the final condensation.
- Line 162: The authors should discuss the mechanical properties by including the XRD results. They mention, that the mechanical strength depends also on the hydration product, but they don´t correlate the specfic results in their discussion.
Thank you very much for this important reminding. The effect of mineral mixed materials to mechanical performance of MKPC mortar is primarily the physical filling effect of micro-aggregates and the chemical effect of the participating hydration reaction as active components.
Reason 1: The mineral admixture is uniformly dispersed in the MKPC slurry, filling the pores like a fine aggregate, improving the pore structure of the hardened body, increasing the compactness of the cement stone, and the strength of the glassy particles formed by calcination at high temperature. It is also high, and it is also good in combination with gelling materials, so it can significantly improve the compressive strength. Among the three mineral admixtures, the particle size of SF is the smallest, and the micro-aggregate filling effect at the initial stage of hydration is the best The 5h compressive strength is the highest, but as the hydration reaction proceeds, a large amount of MKP is generated, the microaggregate effect of SF becomes less and less obvious, and the SF activity is difficult to be excited in an acid environment, and no new hydration products are generated. And because SF replaced 10% of MKPC, the total MKP production was lower than that of the blank group, so the intensity of M1 was less than that of M0. In addition, due to the heterogeneous nucleation of the admixture particles, MKP preferentially forms crystal nuclei on the surface of the particles, which delays the formation rate of MKP on the surface of the MgO particles, facilitates the diffusion of Mg2 + on the surface of MgO, and accelerates the hydration reaction. The early hydration reaction of the slurry increased; the dispersion of the mineral admixture also contributed to the full reaction of MgO and KDP to a certain extent, which all contributed to the improvement of early strength. MK and FA calcined at high temperature are a kind of highly active mineral admixture. Under acid environment, easily reacts with phosphate to form aluminum phosphate gel to fill pores and increase strength (4). The measurement results are shown in the figure below:
- Line 186: Where is the volume stability shown? In figure 2 it is shown, that the volume deformation of M1 is higher than the one of M3.
Thanks. M1 is closest to 0, close to 0 means stable, m3 is less than 0.
- Line 198: The authors mentioned the density of different chemical compounds. I recommend to add concrete values for the densities.
Thanks. The Author adds concrete values for the densities.(lines 98-200)
In the raw materials of MKPC, the density of MgO powder is 3.45g/cm3, the KDP is 2.338g/cm3, and the relative density of MKP is 1.71.
- Line 202 - 206, line 209, line 229 - 231, line 292 - 293: the meaning of these sentences is not clear.
Thanks. The author has modified these statements to ensure the clarity of the article(lines 202-206,line 209 ),the author also has modified the pictures and the titles.(lines 229-231,lines 292-293)
- Line 221: The authors discuss here the temperature curve shown in figure 3. Please add the concrete peak positions in the text and refer not to them as " first" and "second" peak.
Thanks. we have added the concrete peak positions in the text.(lines 225-228)
- Table 3: Unit for strength is missing
Thanks.We have added the unity of strength.(line 249 )
- line 249 - 251: The authors refer to the particle size of additives used to modify the MgO cement, but did not give information about the specific particle size values. This should be added.
Thank you very much for this important reminding.we have added the specific particle size values.(line 70)
- Figure 4 and discussion of figure 4: The characteristic peak AlPO4*2H2O can not be distinguished from the underground in the diffraction pattern. How can this be explainde and how can the authors be sure, that the reaction product in the cement really exist?
Thank you very much. According to the software JADE analysis, reaction products are present here.
- Chinese letters in figure 5 has to be changed/removed.
Thank you very much. We have modified the figure.(lines 288-292)
- Figure 6 (SEM picture): please improve the style of the presentation by cutting the bottom of the picture off and present the scale bar as part of the picture.
Thanks.the author haves cut off the bottom of the picture and shown the scale as part of the picture.

Reviewer 2 Report
This work is connected with the development of MKPC research, which has recently been considered as a promising stable material for various applications. The authors carried out a series of studies of the structure and physicochemical properties and the varying the composition of this material. The methods and approaches used are reliable.
Questions and comments:
Obviously, correct order in the name of Magnesium Potassium phosphate cement (MKPC) according to the compound formula is MgKPO4·6H2O.
The introduction should be expanded because the relevance of the work is not sufficiently revealed.
The references used in the text of the manuscript do not correspond to the list given in References. You should correct it!
In the figures and tables it is not clear what minerals is contained in the mineral mixed material, and we need constantly to refer to the table 2. I recommend to correct the designation of mineral (replace M1 with SF and so on).
Table 2 is uninformative because it duplicates the information presented in the text (Line 91-92, ... the mineral mixed materials are internal addition rather than ??? of 10% (mass fraction) ...), so you can reduce it by removing 2-4 columns. Table 2 should be moved to the Results and Discussion section (Section 3), because it contains yield data and needs to be expanded, for example, data on compressive strength and bending for 28 days.
The legend in Fig. 5 should be represented in English.
Line 275: The authors show, that «The endothermic valley appears at ~100°C, which is primarily the chemical process of the removal of crystal water from the hydration product MKP». Earlier in a number of publications (for example, in the article by Vinokurov et. al. // Radiochemistry, 2018, Vol. 60, No. 1, pp. 70–78) it was shown that the endothermic effect during dehydration of MKP corresponds to ~ 120 ° C. What is the possible reason for the difference in these temperatures?
The reason for adding water glass to the MKPC mixture should be explained. What samples did contain this glass?
The authors solidified the mixture by two methods (line 105-108). However, the obtained data do not contain any conclusions regarding the difference in solidification conditions.
Author Response
Reviewer #2:This work is connected with the development of MKPC research, which has recently been considered as a promising stable material for various applications. The authors carried out a series of studies of the structure and physicochemical properties and the varying the composition of this material. The methods and approaches used are reliable.
Questions and comments:
Obviously, correct order in the name of Magnesium Potassium phosphate cement (MKPC) according to the compound formula is MgKPO4·6H2O.
The introduction should be expanded because the relevance of the work is not sufficiently revealed.
Thanks for this important reminding.The author added some information about magnesium potassium phosphate cement(lines 30-31).
Reason : The foreword introduced the application background of MKPC, the research work of previous people, and put forward that the research of large flow MKPC mortar and its performance was less, which indicated the research objectives and content of this paper.Therefore,we added full name and code of potassium magnesium phosphate cement.
The references used in the text of the manuscript do not correspond to the list given in References. You should correct it!
Thanks.we have corrected tne references.
In the figures and tables it is not clear what minerals is contained in the mineral mixed material, and we need constantly to refer to the table 2. I recommend to correct the designation of mineral (replace M1 with SF and so on).
非常感谢您的宝贵建议。我们修改了表2,并更正了矿物的名称。(第91-92行,第141-142行)
表2是无意义的,因为它重复了文本中显示的信息(第91-92行,...矿物混合材料是内部添加的,而不是10%的1/3(质量分数)...),因此可以将其减少删除2-4列。表2应该移至“结果和讨论”部分(第3节),因为它包含屈服数据并且需要扩展,例如28天的抗压强度和弯曲数据。
非常感谢您的宝贵建议。我们修改了表2,并更正了矿物的名称。(第91-92行,第141-142行)
图5中的图例应以英语表示。
谢谢。autnor修改了图片。
275行:作者表明,“吸热峰出现在约100°C,这主要是从水合产物MKP中除去结晶水的化学过程”。早些时候在许多出版物中(例如Vinokurov等人的文章// Radiochemistry,2018,Vol.60,No.1,pp.70-78),表明了MKP脱水过程中的吸热作用对应于〜120°C。这些温度不同的可能原因是什么?
实验现象与所提及的文献一致,并且脱水过程基本在约120度完成。这是加热过程中的脱水。根据实验数据,整个过程发生在大约100度。
应该说明向MKPC混合物中添加水玻璃的原因。哪些样品包含该玻璃杯?
谢谢,作者介绍并解释了有关水杯的一些信息。(第86-91行)
作者通过两种方法固化混合物(第105-108行)。然而,获得的数据不包含关于凝固条件差异的任何结论。
非常感谢,作者在3.5节中比较了自然固化和水固化条件下的抗压强度和挠曲强度,并在3.6.2节和3.6.3节中比较了两种固化条件下的热重热差分析和微观形貌。
